# Effects of Self-Efficacy, Depression, and Anger on Health-Promoting Behaviors of Korean Elderly Women with Hypertension

**DOI:** 10.3390/ijerph17176296

**Published:** 2020-08-29

**Authors:** Ae Sil Kim, Mi Heui Jang, Kyung Hwan Park, Ji Young Min

**Affiliations:** College of Nursing Science, Kyung Hee University 26, Kyungheedae-ro, Dongdaemun-gu, Seoul 02447, Korea; interkim2@hanmail.net (A.S.K.); siloam02@naver.com (K.H.P.); freemjy@chu.ac.kr (J.Y.M.)

**Keywords:** elderly, women, health promotion, hypertension, anger

## Abstract

The prevalence of hypertension among women in Korea aged 65 years or older is 61.7%. Past research has emphasized the importance of health-promoting behaviors in hypertension management for the elderly. The purpose of this cross-sectional study was to identify the effects of self-efficacy, depression, trait anger, and anger expression on the health-promoting behaviors of elderly women with hypertension. Self-report questionnaires were completed by 208 women aged 65 and older (age range: 65 to 85) diagnosed with hypertension by physicians and living in the communities of G city and N city of Gyeonggi-do in South Korea. A hierarchical regression analysis revealed that exercise (β = 0.36, *p* < 0.001) had the most significant effect on health-promoting behaviors, followed by depression (β = −0.31, *p* < 0.001), trait anger (β = 0.21, *p* = 0.002), anger control (β = 0.20, *p* < 0.001), religion (β = 0.18, *p* = 0.001), and self-efficacy (β = 0.18, *p* = 0.003). Trait anger and anger control were identified to have a more significant effect on health-promoting behaviors than self-efficacy. Thus, health-promoting behaviors were influenced by exercise, depression, anger, religion, and self-efficacy. It is necessary to implement a nursing intervention strategy which pays attention to these factors to improve health-promoting behaviors of Korean community-dwelling elderly women.

## 1. Introduction

Hypertension is a major cause of early deaths worldwide [1]. It accounts for the largest proportion of medical costs among all diseases in Korea owing to a rapid increase in the elderly population with the prevalence of hypertension among women aged 65 years or older reported to be 61.7% [2]. Hypertension leads to complications such as heart diseases and cerebrovascular diseases, which eventually cause death.

The rates of recognition, treatment, and control of hypertension in Korean patients with hypertension are 65.0%, 61.0%, and 44.0% respectively [3]; the rate of hypertension control in Korea is currently low. A total of 46.0% patients with hypertension in Korea were aged 65 years or older [4]. The low rate of hypertension control among patients is attributed to the difficulty of hypertension management due to the asymptomatic nature of the disease [4]. It is necessary to help the elderly engage in health-promoting behaviors to consistently control their blood pressure. Switching to a healthier lifestyle is necessary for hypertension management in the elderly. The importance of a preventive approach that encourages the elderly to engage in health-promoting behaviors and strictly and proactively manage their health has been emphasized [5].

A previous study on hypertension in the elderly [6], emphasized that it is important that the elderly consistently engage in health-promoting behaviors in addition to getting a regular medical check-up and taking medications, as their blood pressure is not effectively controlled by regular medical check-ups and consistent use of antihypertensive medications. Health-promoting behaviors refer to the act of proactively responding to the environment to achieve a high health status. They include all aspects of a healthy lifestyle, including physical activities, responsibility for health, spiritual growth, diet, interpersonal relationships, and stress management all aimed at promoting one’s well-being, self-realization, and self-achievements [7].

Previous studies [8,9,10,11,12,13,14,15,16,17,18,19] have reported that health-promoting behaviors are associated with self-efficacy in the elderly. Self-efficacy refers to an individual’s belief in their ability to successfully perform a task, and is a major cognitive factor that encourages an individual to continuously engage in health-promoting behaviors [4]. Previous studies [9,10], have reported that self-efficacy is an important factor that promotes the elderly to manage and control their blood pressure, and is highly associated with health-promoting behaviors. It has also been consistently reported that depression has a significant effect on the health-promoting behaviors of elderly people with hypertension [11,12,13,14]. The rate of engaging in self-nursing behaviors decreased as the level of depression increased in elderly women with hypertension who had low income [12,13], and the rate of engaging in health-promoting behaviors decreased as the level of depression increased in the elderly living in local communities [14]. The rate of depression has been reported to be 12.0% higher for elderly women than elderly men in Korea who were examined for geriatric depression [15], indicating that elderly women are more susceptible to depression than elderly men. Therefore, it is necessary to examine and assess depression in elderly women to prevent any negative outcomes of depression related to health-promoting behaviors.

Researchers have reported that cardiovascular diseases such as hypertension are closely associated with the human emotion of anger, and that psychological factors such as anger can affect the rate of morbidity and mortality of cardiovascular diseases [16,17]. Anger is an important psychological trait in the elderly [18], and is a universal human emotion and a natural reaction to various events that inevitably happen in daily life [19,20]. Trait anger is a dispositional characteristic of an individual that can express state anger [21]. An anger expression refers to a behavioral response of an individual based on the feelings of anger experienced. There are three anger expression styles: anger-in, anger-out, and anger control. Anger-in is suppressed and internalized anger, and anger-out is externalized anger expressed toward an object or another person. Anger control is the attempt to appropriately control one′s anger expression [22]. Koreans are culturally oriented by Confucianism to respect others and suppress their emotions while communicating with others [22]. Hwabyeong, which is a Korean cultural syndrome, has been reported to be highly associated with angry emotions. Those with hwabyeong tend to suppress negative emotions and focus on their inner selves. Due to these cultural characteristics, Koreans have been reported to commonly use the anger-in expression style, meaning they tend to hold in, suppress, and internalize the feelings of anger [23,24,25]. Anger suppression can lead to the physical manifestations of hypertension and hwabyeong, thereby significantly negatively affecting health. Therefore, it is necessary to examine the association between hypertension, for which appropriate blood pressure control is crucial, and health-promoting behaviors in elderly women. Blood pressure control is especially important for elderly women as they are susceptible to cardiovascular problems due to postmenopausal reduction in the level of estrogen, a female sex hormone [26,27].

Most studies that examined the health-promoting behaviors of community-dwelling elderly with hypertension [28,29,30], analyzed elderly men and women as a single group and did not consider gender differences. In Korean culture, anger is an important psychological factor that affects female health; thus, it is necessary to additionally examine the effect of anger on health-promoting behaviors [17]. The conceptual framework of this study was established as shown in Figure 1, based on the review of relevant literature. The aim of this study was to investigate the effect of self-efficacy, depression, trait anger, and anger expression styles on the health-promoting behaviors of elderly women with hypertension in local communities.

## 2. Materials and Methods

### 2.1. Study Design and Participants

This cross-sectional study investigates the effects of exercise, self-efficacy, depression, trait anger, and anger expression on the health-promoting behaviors of elderly women with hypertension living in local communities. Participants were women aged 65 years or older (age range: 65 to 85) living at home in local communities in G city and N city in Gyeonggi-do, South Korea, who were diagnosed with hypertension. The sample size was calculated using the sample size calculator in G*Power with the power at 0.95, effect size at medium 0.15, significance level at 0.05, and number of predictors at 16. The sample size needed for multiple regression analysis was calculated at 204. Considering the possibility of incomplete questionnaire responses, 220 participants were included. Of the 220 questionnaires collected, 12 were deemed to have inadequate data, owing to withdrawal of participation while completing the questionnaire. These were excluded, and the remaining 208 questionnaires were used in this study (Figure 2).

The inclusion criteria used in this study were: (1) being aged 65 years or older and diagnosed with hypertension by a doctor; (2) being able to listen and respond to questionnaire questions; and (3) understanding of the purpose of the study and provision of consent to participate. The exclusion criteria were: (1) having cognitive problems such as dementia; (2) diagnosis of a lethal disease such as cancer; and (3) inability to communicate.

### 2.2. Measures

#### 2.2.1. Sociodemographic Characteristics and Health-Related Characteristics

Demographic and health-related characteristics including age, education, religion, monthly income, perceived health status, exercise, smoking status, alcohol consumption, use of antihypertensive medications, and previous experience with hypertension education, were investigated. 

#### 2.2.2. Self-Efficacy 

A self-efficacy assessment tool for patients with hypertension developed by Park [31] was used. Scores for each question range from 10 for “not confident at all” to 100 points “totally confident”. Higher scores indicate higher self-efficacy. For convenience, the scores were modified to range from 1 to 10 points in this study. The scale had a Cronbach′s α of 0.72 at the time of its development and 0.78 in this study.

#### 2.2.3. Depression 

The Geriatric Depression Scale Short Form Korea Version (GDSSFK) by Kee [32], which is a version of the Geriatric Depression Scale (GDS) developed by Ysavage and Sheikh [33], that was adapted for the Korean elderly, was used to measure depression. It consists of 15 questions. Total scores of 0–5 points indicate that a participant is normal, and 6 points or higher indicate depression. Each question is answered with either “yes” or “‘no” with 0 and 1 point allocated for the respective responses [32]. Items 2, 7, 8, 11, and 12 were negatively worded questions and were reverse scored. The scale had a Cronbach′s α of 0.88 in a study by Kee [32], and 0.79 in this study.

#### 2.2.4. Trait Anger and Anger Expression 

The Korean version of the State-Trait Anger Expression Inventory (STAXI-K) [34], adapted from the State-Trait Anger Expression Inventory developed by Spielberger [21], and translated into Korean was used to measure anger. Trait anger refers to a dispositional characteristic of an individual that can express state anger. The tool consists of 10 questions rated on a four-point Likert scale; 1 point is allocated for “not true at all”, 2 points for “somewhat true”, 3 points for “true”, and 4 points for “very true”. The total score ranges from 10 to 40 points. Higher scores for trait anger indicate higher levels of perceived anger. The tool has a total of 24 questions rated on a four-point Likert scale; 1 point is allocated for “not true at all”, 2 points for “somewhat true”, 3 points for “true”, and 4 points for “very true”. Total scores range from 8 to 32 points. Higher scores on a certain anger expression style indicate a higher rate of using that style. The tool had a Cronbach′s α of 0.83 for trait anger, 0.65 for anger-in, 0.69 for anger-out, and 0.79 for anger control at the time of its development [34], and values of 0.86, 0.76, 0.74, and 0.85 respectively in this study.

#### 2.2.5. Health-Promoting Behaviors 

A Korean adaptation of the Health Promoting Lifestyle Profile II (HPLP-II) developed by Walker et al. [35], translated and validated by Yun and Kim [36], was used to assess health-promoting behaviors. The HPLP-II consists of 6 domains with a total of 52 questions: physical activity (8 questions), responsibility for health (9 questions), spiritual growth (9 questions), diet (9 questions), interpersonal relationship (9 questions), and stress management (8 questions). Each question is rated on a four-point Likert scale; 1 point is allocated for “not at all”, 2 points for “sometimes true”, 3 points for “often true”, and 4 points for “always true”. Total scores range from 52 to 208 points. Higher scores indicate higher levels of engagement in health-promoting behaviors. The original scale had a Cronbach′s α of 0.94 at the time of its development, and the version by Yun and Kim [36], had a Cronbach′s α of 0.91. The Cronbach′s α was 0.91 in this study.

### 2.3. Procedure

For data collection, the researchers informed the heads of a community center in G city and senior community centers in G city and N city in Gyeonggi-do of the purpose of this study and received permission to conduct research at the centers. The researchers visited each center to collect data. The researchers and research assistants met participants in a consultation room provided by the community center. The researchers and research assistants remained with the participants and helped them fill out the questionnaires in a quiet and undisturbed space. Self-reported questionnaires were used and instructions were provided on how to complete them. Participants completed the questionnaire using a pen on their own if they were able to do so. A researcher then reviewed and collected the completed questionnaires. For participants who had literacy difficulties, mobility problems, or poor vision, a researcher read the questions aloud and a research assistant recorded the participants’ responses to the questions. All data were collected by a single researcher and a single research assistant. The questionnaire took approximately 30 min to complete. Participants were compensated with wet wipes for completing the questionnaire. Blood pressure was measured before the survey; participants were asked not to smoke or ingest caffeine-containing food or drinks such as coffee 30 min before the measurement. Blood pressure was measured using an automatic digital blood pressure reader (OMRON HEM-181 7111; Omron Healthcare Co., LTD., Kyoto, Japan), and the participants were requested to sit and raise their arm up to their chest for the measurements. After the participants rested, two measurements were obtained at a 10-min interval.

### 2.4. Statistical Analysis

Descriptive statistics, including error, percentage, mean, and standard deviation, were analyzed for the demographic and health-related characteristics. Differences in health-promoting behaviors according to demographic and health-related characteristics were analyzed using a t-test or F-test. Pearson′s correlation coefficients were calculated to determine the correlations between major factors. A hierarchical multiple regression analysis was performed to analyze the effects of different factors on health-promoting behaviors. Nominal independent variables were represented by dummy variables. SPSS 23.0 was used for statistical analysis.

### 2.5. Ethical Considerations

This study was approved by the Institutional Review Board (IRB) of K University (Approval number: KHSIRB-14-037(RA)). All participants were informed about the content and purpose of this study and provided a consent form without a signature before the start of the study.

## 3. Results

### 3.1. Demographic and Health-Related Characteristics

Table 1 shows the demographic and health-related characteristics of the participants. A total of 116 participants (55.8%) were aged 70–79 years, accounting for the largest proportion; 135 participants (64.9%) had completed elementary school education or less. Most participants (n = 178, 85.6%) had a monthly income <833.3 $, and 30 participants (14.4%) had a monthly income ≥833.3 $. The participants′ mean systolic blood pressure was 130 mmHg, and the mean diastolic pressure was 75 mmHg. The maximum and minimum blood pressure measured was 178 mmHg and 53 mmHg, respectively. Eighty participants (38.5%) reported their health as “poor”, and 152 (73.1%) said they “exercised regularly”. Smoking status, alcohol consumption, current or past history of taking antihypertensive agents, and previous experience with hypertension education were also investigated.

### 3.2. Differences in Health-Promoting Behaviors According to Demographic and Health-Related Characteristics

No significant differences in the level of engagement in health-promoting behaviors were found according to most of the demographic variables, excluding religion. The level of engagement in health-promoting behaviors was significantly higher for participants with a religion compared to those without (129.67 ± 21.07 and 122.34 ± 22.07, respectively; t = −1.98, *p* = 0.049). With respect to health-related characteristics, significant differences in the level of engagement in health-promoting behaviors were found only for regular exercise (t = 8.82, *p* < 0.001) (Table 2).

### 3.3. Effects of Self-Efficacy, Depression, Trait Anger, and Anger Expression on Health-Promoting Behaviors

The mean scores were 128.23 ± 21.42 (range: 82–182 points) for health-promoting behaviors, 62.60 ± 9.28 (range: 28–80 points) for self-efficacy, 5.03 ± 3.30 (range: 0–14 points) for depression, and 19.12 ± 5.65 points (range: 10–38 points) for trait anger. The mean scores for anger-in, anger-out, and anger control were 13.25 ± 3.86 points (range: 8–25 points), 13.63 ± 4.01 points (range: 8–27 points), and 20.75 ± 4.95 points (range: 9–32 points), respectively (Table 3).

### 3.4. Correlations of Health-Promoting Behaviors with Self-efficacy, Depression, Trait Anger, and Anger Expression

Self-efficacy was significantly positively correlated (r = 0.514, *p* < 0.001), depression was significantly negatively correlated (r = −0.439, *p* < 0.001), and anger-control was significantly positively correlated (r = 0.230, *p* < 0.001) with health-promoting behaviors. However, no significant correlations were found between trait anger, anger-in, and anger-out and health-promoting behaviors (Table 4).

### 3.5. Factors Influencing Health-Promoting Behaviors

A hierarchical multiple regression analysis was performed to examine the amount of variance in health-promoting behaviors explained by the independent variables (Table 5). In a multicollinearity test, no extreme coefficient values ≥0.8 were found between the independent variables, indicating a low risk of multicollinearity. All independent variables had variance inflation factors ≤10 and tolerance ≥0.1, indicating no presence of multicollinearity. The final model in this study had F = 25.49, *p* < 0.001 and was deemed to have a good fit for the data. In the four-stage research model of hierarchical regression analysis, the order of inputting independent variables in this study was as follows: individual characteristics, cognitive factors, and emotional factors. In particular, self-efficacy and depression, which are well known as factors affecting health-promoting behaviors among the elderly, were inputted first, and anger was inputted last in order to examine the effects of anger on health-promoting behavior. In Model 1, among the sociodemographic and health-related characteristics, religion and exercise (which showed significant associations with health-promoting behaviors) were entered as independent variables into the regression model. Self-efficacy was additionally entered into Model 2, as was depression in Model 3 and trait anger, anger-in, anger-out, and anger control in Model 4.

The hierarchical multiple regression analysis was performed in four stages. The regression equations of Models 1 to 4 were statistically significant (F = 40.38, *p* < 0.001; F = 40.75, *p* < 0.001; F = 41.16, *p* < 0.00; F = 25.49, *p* < 0.001, respectively). The variables added to Model 4 were religion, exercise, self-efficacy, depression, trait anger, anger-in, anger-out, and anger control. Of these, religion, exercise, self-efficacy, depression, trait anger, and anger control explained a significant amount of variance (49.0%) in health-promoting behaviors, which was 6.0% more than the amount of variance explained by Model 3. Exercise (β = 0.36, *p* < 0.001) had the greatest effect on the health-promoting behaviors of elderly women with hypertension, followed by depression (β = -0.31, *p* < 0.001), trait anger (β = 0.21, *p* = 0.002), anger control (β = 0.20, *p* < 0.001), religion (β = 0.18, *p* = 0.001) and self-efficacy (β = 0.18, *p* = 0.003).

## 4. Discussion

This study was conducted to examine the effects of self-efficacy, depression, trait anger, and anger expression styles on the health-promoting behaviors of elderly women with hypertension living in local communities and to provide basic data for the development of a related health promotion program. This study focused on understanding the effect of anger by serially adding variables including self-efficacy, depression, trait anger, and anger expression styles to a regression model and examining the amount of variance in health-promoting behaviors explained by each of these variables. Based on the results of this study, exercise had the highest effect on the health-promoting behaviors of elderly women with hypertension, followed by depression, trait anger, anger control, religion, and self-efficacy.

The effect of exercise found in this study is consistent with the results of previous studies that did not use the same assessment tools but involved healthy elderly women [37,38]. In these studies, elderly women who participated in health gymnastics were found to take good care of their health and have a healthy lifestyle. In a study by Kwon [37], participating in health gymnastics was found to have a positive effect on all domains of health-promoting behaviors in elderly women. Thus, the results of this study reconfirm the important effect of exercise on the health-promoting behaviors of elderly women with hypertension.

In this study, the level of engagement in health-promoting behaviors increased as the level of depression decreased in elderly women with hypertension. This finding is consistent with previous reports that depression is a risk factor in health-promoting behaviors [14,39], and that the level of engagement in health-promoting behaviors increases as the level of depression decreases in the general elderly and elderly women with hypertension who have low income [9,40]. These results demonstrate the need to expand the early screening process for depression and to prevent and take preemptive measures against depression to improve the health-promoting behaviors of elderly women requiring hypertension management. It may also be necessary to develop and implement a program that encourages the elderly to voluntarily engage in health-promoting behaviors to maintain their bodily functions.

Based on the final hierarchical regression model, trait anger, and anger control were found to significantly affect health-promoting behaviors when the effects of self-efficacy and depression were controlled. In other words, elderly women with hypertension who had higher levels of trait anger and anger control had higher levels of engagement in health-promoting behaviors. The association between trait anger and health-promoting behaviors was in contrast to the researchers′ expectations. Trait anger, which refers to the disposition or tendency to express anger based on the perceived frequency and intensity of anger, positively affected health-promoting behaviors in this study. This observation may be attributed to the ambivalent nature of trait anger. Anger is a negative emotion that leads to destructive behavior and at the same time a positive emotion that promotes adrenaline production for survival. As trait anger has both these dysfunctional and functional aspects of anger [41], and in other words, is ambivalent, it is possible that elderly women with high levels of trait anger have good problem-solving skills. Trait anger is a justified expression of anger and a positive emotion related to self-esteem necessary for self-protection and may play a protective role in constructive activities such as problem solving [40].

The positive effects that trait anger and anger control had on health-promoting behaviors can be explained by the function of trait anger and anger control expression [40]. Among the three anger expression styles, anger-out and anger-in are considered dysfunctional, while anger control is considered functional [41,42]. The results of this study support a previous report that anger control is an important mediation variable that changes the health-promoting behaviors of the elderly of both genders with different levels of depression, and that anger control has the highest effect on health-promoting behaviors among the three anger expression styles [40]. Korean elderly women suffer from cultural syndromes such as hwabyeong that result from anger suppression (or anger-in) imposed on them by the patriarchal family culture based on Confucianism, conflicts with younger generations, and financial difficulties [23,24,25], and need a constructive method of anger expression and control. As anger control was found to positively affect the health-promoting behaviors of elderly women with hypertension, it is necessary to devise a nursing intervention strategy that focuses on anger control. A health promotion program involving aspects such as cognitive behavioral therapy and mindfulness-based intervention should be implemented to assist elderly women with hypertension in anger control.

In this study, participants with a religion had higher levels of engagement in health-promoting behaviors than those without [43], consistent with a previous report [43]. Pender [7], explained that spiritual growth allows individuals to set a life goal and undergo positive changes in their lives. It is possible that the participants in this study underwent spiritual growth through religious activities, which positively affected their health-promoting behaviors. Furthermore, the belief that one is connected to an absolute being may have provided psychological relief and positively affected the lives of the elderly.

In this study, participants with higher levels of self-efficacy were found to have higher levels of engagement in health-promoting behaviors, consistent with previous findings [7,44]. This indicates the necessity of providing a nursing intervention to improve the self-efficacy of elderly women with hypertension who must be encouraged to engage in health-promoting behaviors. A noteworthy result of this study is that in the final model wherein anger was added, the effect of depression increased while that of self-efficacy decreased. Considering how trait anger and anger control had a relatively higher effect on health-promoting behaviors than self-efficacy, the need to develop and implement an anger management program for elderly women with hypertension in Korea is even clearer.

In summary, increasing physical activities and early screening and management of depression in elderly women with hypertension is necessary to increase their engagement in health-promoting behaviors. It is also necessary to develop a health promotion program with a nursing intervention strategy aimed at improving the appropriate anger control skills and self-efficacy of elderly women with hypertension in local communities. The importance of self-efficacy and depression has been consistently emphasized in the development of a health promotion program for the elderly. However, there is inadequate research on anger, which is a psychological factor that has a direct effect on the health behaviors of elderly women with hypertension, requiring lifelong care. This study is meaningful in that it identified trait anger and anger control as the major psychological factors affecting the health-promoting behaviors of elderly women with hypertension and demonstrated the need to pay attention to anger as an important factor affecting health-promoting behaviors.

The results of this study may be used as basic data for the development of an integrated health promotion program for elderly women with hypertension that aims to improve their anger control and management skills by considering not only factors such as exercise, depression, and self-efficacy but also anger, which is a psychological trait.

This study has some limitations. First, it was difficult to generalize the results because the participants were Korean elderly women with hypertension from specific regions, and different results might occur depending on culture and ethnicity. Second, most participants in this study were taking antihypertensive medications and controlled their blood pressure in the normal range. Therefore, we suggest that further research is needed, including with participants who experience difficulties in controlling their blood pressure.

## 5. Conclusions

The effects of demographic and health-related characteristics, self-efficacy, depression, trait anger, and anger expression on the health-promoting behaviors of elderly women with hypertension were analyzed in this study. Exercise was found to have the greatest effect on health-promoting behaviors, followed by depression, trait anger, anger control, religion, and self-efficacy. Thus, by enhancing these factors it may be possible to increase the engagement of elderly women with hypertension in health-promoting behaviors.

## Figures and Tables

**Figure 1 ijerph-17-06296-f001:**
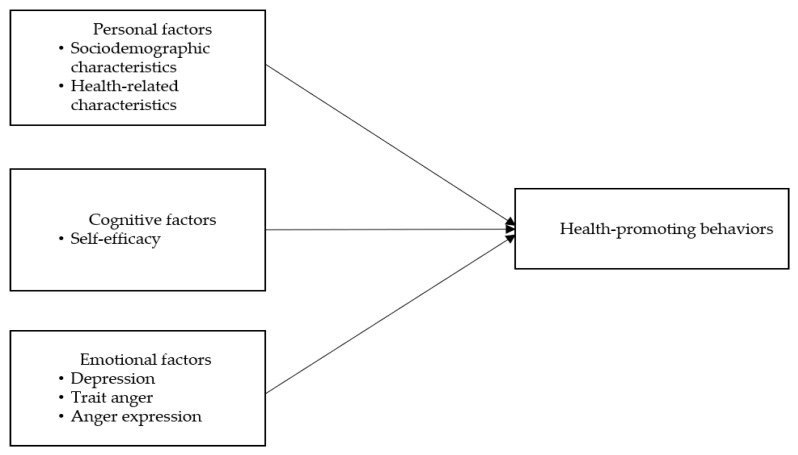
The conceptual framework.

**Figure 2 ijerph-17-06296-f002:**
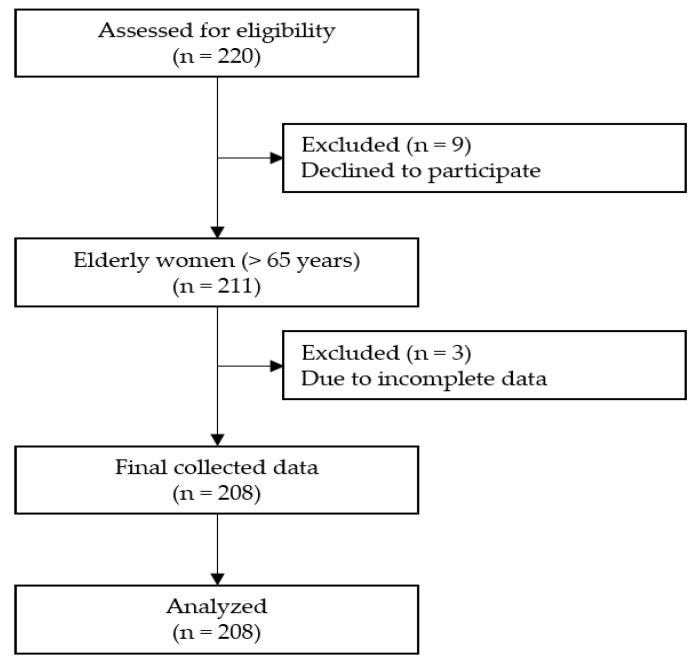
Flowchart of the study.

**Table 1 ijerph-17-06296-t001:** Sociodemographic and health-related characteristics of participants (N = 208).

Characteristics	Categories	N	%
Age (years) Mean (SD) = 75.7 (6.09)	65–69	35	16.8
	70–79	116	55.8
	≥80	57	27.4
Education	≤Elementary school	135	64.9
	≥Middle school	73	35.1
Religion	Yes	167	80.3
	No	41	19.7
Monthly income (US dollars) Mean (SD) = 1043.7 (0.64)	<833.3 $	178	85.6
	≥833.3 $	30	14.4
Perceived health status	Bad	80	38.5
	Moderate	98	47.1
	Good	30	14.4
Regular exercise (3 times/week)	Yes	152	73.1
	No	56	26.9
Smoking	Yes	5	2.4
	No	203	97.6
Drinking	Yes	39	18.8
	No	169	81.3
Hypertension drugconsumption	None	14	6.7
	Regular	187	89.9
	Irregular	7	3.4
Hypertension educational experience	Yes	62	29.8
	No	146	70.2

SD = standard deviation.

**Table 2 ijerph-17-06296-t002:** Differences in health-promoting behaviors according to sociodemographic.

Characteristics	Categories	M ±SD	t or F(df)
Age (years) Mean (SD) = 75.7 (6.09)	65–69	133.11 ± 25.33	1.42 (2, 205)
	70–79	126.33 ± 20.73	
	≥80	129.09 ± 20.00	
Education	≤Elementary school	127.28 ± 21.63	−0.85 (207)
	≥Middle school	129.97 ± 21.06	
Religion	Yes	129.67 ± 21.07	−1.98 (207) *
	No	122.34 ± 22.07	
Monthly income (US dollars) Mean (SD) = 1043.7 (0.64)	<$833.3	127.69 ± 20.59	−0.74 (207)
	≥$833.3	131.40 ± 25.95	
Perceived health status	Bad	124.64 ± 22.64	
	Moderate	129.08 ± 20.68	2.74 (2, 205)
	Good	135.00 ± 19.03	
Regular exercise (3 times/week)	Yes	132.84 ± 18.69	8.82 (207) ***
	No	108.02 ± 15.92	
Smoking	Yes	115.60 ± 23.37	−1.34 (207)
	No	128.54 ± 21.33	
Drinking	Yes	133.74 ± 20.57	−1.80 (207)
	No	126.95 ± 21.47	
Hypertension drug consumption	None	119.79 ± 19.71	2.63(2, 205)
	Regular	129.34 ± 21.39	
	Irregular	115.43 ± 19.97	
Hypertension educational experience	Yes	132.18 ± 21.42	−1.72 (207)
	No	126.61 ± 21.32	

* *p* < 0.05, *** *p* < 0.001; SD = standard deviation; df = degree of freedom.

**Table 3 ijerph-17-06296-t003:** Descriptive statistics and measured range of variables.

Variables	Items	M ± SD	Minimum	Maximum	Range
Health-promoting behaviors	52	128.23 ± 21.42 (2.02 ± 1.02)	82	182	52-208
Physical activity	8	18.25 ± 6.99 (2.28 ± 0.87)	8	32	8-32
Responsibility for health	9	21.43 ± 4.77 (2.38 ± 0.53)	10	34	9-36
Spiritual growth	9	20.10 ± 5.13 (2.23 ± 0.57)	10	35	9-36
Diet	9	25.07 ± 4.12 (3.13 ± 0.51)	13	35	9−36
Interpersonal relationships	9	22.72 ± 4.90 (2.52 ± 0.54)	11	34	9−36
Stress management	8	20.47 ± 3.97 (2.27 ± 0.44)	10	31	8−32
Self-efficacy	10	62.60 ± 9.28 (7.58 ± 1.03)	28	80	1−100
Depression	15	5.03 ± 3.30 (0.34 ± 0.22)	0	14	0−15
Trait anger	10	19.12 ± 5.65 (1.91 ± 0.26)	10	38	10−40
Anger-in	8	13.25 ± 3.86 (1.66 ± 0.48)	8	25	8−32
Anger-out	8	13.63 ± 4.01 (1.70 ± 0.50)	8	27	8−32
Anger control	8	20.75 ± 4.95 (2.59 ± 0.62)	9	32	8−32

**Table 4 ijerph-17-06296-t004:** Correlation among variables.

Variables	X1	X2	X3	X4	X5	X6	Y1
X1 Self-efficacy	1						
X2 Depression	−0.296 ***	1					
X3 Trait anger	−0.002	0.099	1				
X4 Anger-in	0.074	0.256 ***	0.374 ***	1			
X5 Anger-out	−0.002	−0.009	0.582 ***	0.268 ***	1		
X6 Anger control	0.222 ***	−0.009	−0.064	0.205 **	−0.113	1	
Y Y1 Health-promoting behaviors	0.514 ***	−0.439 ***	0.084	0.006	0.066	0.230 ***	1

** *p* < 0.01, *** *p* < 0.001.

**Table 5 ijerph-17-06296-t005:** Hierarchical multiple regression analysis of factors influencing health-promoting behaviors (N = 208).

Model.	Model 1				Model 2				Model 3				Model 4			
Variables	β	*p*	Multicollinearity		Β	*p*	Multicollinearity		β	*p*	Multicollinearity		β	*p*	Multicollinearity	
			TOL	VIF			TOL	VIF			TOL	VIF			TOL	VIF
Religion *	0.11	0.067	0.997	1.003	0.11	0.043	0.997	1.003	0.14	0.008	0.987	1.013	0.18	0.001	0.953	1.050
Exercise *	0.51	<0.001	0.997	1.003	0.35	<0.001	0.769	1.300	0.32	<0.001	0.763	1.311	0.36	<0.001	0.724	1.380
Self-efficacy					0.35	<0.001	0.771	1.296	0.08	<0.001	0.719	1.391	0.18	0.003	0.634	1.576
Depression									−0.29	<0.001	0.873	1.145	−0.31	<0.001	0.785	1.274
Trait anger													0.21	0.002	0.587	1.704
Anger-in													−0.08	0.167	0.720	1.389
Anger-out													−0.02	0.771	0.641	1.560
Anger control													0.20	<0.001	0.814	1.229
R^2^	0.28				0.37				0.45				0.51			
Adjusted R^2^	0.28				0.37				0.44				0.49			
F	40.38 (<0.001)				40.75 (<0.001)				41.16 (<0.001)				25.49 (<0.001)			
R^2^ change					0.09 (<0.001)				0.07 (<0.001)				0.06 (<0.001)			

* Dummy variables: Religion (0 = No, 1 = Yes), Exercise (0 = Never exercise, 1 = Regular exercise); ß=Standardized beta; TOL = tolerance; VIF = variance inflation factor.

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
