# Peer review of "Effects of Self-Efficacy, Depression, and Anger on Health-Promoting Behaviors of Korean Elderly Women with Hypertension"

_ijerph, 2020, doi:10.3390/ijerph17176296_

Round 1

Reviewer 1 Report

The topic of this study is interesting. I only have several minor comments.

  1. "This cross-sectional study investigates the effects of self-efficacy, depression, trait anger, and 100 anger expression on the health-promoting behaviors of elderly women with hypertension living in 101 local communities. " (in 2.1. Study Design and Participants) Exercise was not mentioned in the study design.
  2. "The 105 sample size needed for multiple regression analysis was calculated at 146. Considering the possibility 106 of incomplete questionnaire responses, 220 participants were included." (in 2.1. Study Design and Participants) Could the authors please clarify the relationship between the numbers 146 and 220? What is the hypothesized participant drop-off rate?
  3. "Most participants (n = 178, 201 85.6%) had a monthly income <$1,700, and 30 participants (14.4%) had a monthly income ≥$1,700." (in 3.1. Demographic and Health-related Characteristics). Could the authors provide rationale about why $1,700 was used as the threshold? Was it US dollar?
  4. Statistical analyses: Could the authors please also report standard deviation (together with mean) and degree of freedom (e.g., together with t value and p value)? In addition, it is unclear what are the four models why and how the authors selected these four models, and how they compared the four models to decide the best model.

Author Response

The authors would like to thank the reviewers for carefully reviewing our manuscript and providing us Point and suggestions to improve the quality of the manuscript. We have tried our best to address all the reviewers’ concerns, and we will be happy to provide any further answers as needed.

Response to Reviewer 1 Comments

Point 1: "This cross-sectional study investigates the effects of self-efficacy, depression, trait anger, and 100 anger expression on the health-promoting behaviors of elderly women with hypertension living in 101 local communities. " (in 2.1. Study Design and Participants) Exercise was not mentioned in the study design.

Response: We added “exercise” to the description of the study design.

Point 2: "The 105 sample size needed for multiple regression analysis was calculated at 146. Considering the possibility 106 of incomplete questionnaire responses, 220 participants were included." (in 2.1. Study Design and Participants) Could the authors please clarify the relationship between the numbers 146 and 220? What is the hypothesized participant drop-off rate?

Response: Thank you for your comment. There was an error in the description of the sample size. The number of predictors was 16 and the sample size needed was 204. Considering an anticipated dropout rate of less than 10%, 220 participants were included.

Point 3: "Most participants (n = 178, 201 85.6%) had a monthly income <$1,700, and 30 participants (14.4%) had a monthly income ≥$1,700." (in 3.1. Demographic and Health-related Characteristics). Could the authors provide rationale about why $1,700 was used as the threshold? Was it US dollar?

Response: Thank you for pointing this out. There was an error in converting KRW to USD. The minimum monthly cost of living  per person in Korea is about 1,000,000 KRW, which is approximately equivalent to 833.3 USD. We revised the sentence and values in Table 1 accordingly.

Point 4: Statistical analyses: Could the authors please also report standard deviation (together with mean) and degree of freedom (e.g., together with t value and p value)? In addition, it is unclear what are the four models why and how the authors selected these four models, and how they compared the four models to decide the best model.

Response: Thank you for pointing out this important issue. We provided mean and SD values in Table 1 and degree of freedom values in Table 2. In addition, we presented the reason why the four models were selected by stage, and included a detailed description of the models. The suitability of the final model was presented using statistical values.

All independent variables had variance inflation factors ≤10 and tolerance ≥0.1, indicating no presence of multicollinearity. The final model in this study had F = 25.49, p < 0.001 and was deemed to have a good fit for the data.  In the four-stage research model of hierarchical regression analysis, the order of inputting independent variables in this study was as follows: individual characteristics, cognitive factors, and emotional factors. In particular, self-efficacy and depression, which are well known as factors affecting health-promoting behaviors among the elderly, were inputted first and anger was inputted last in order to examine the effects of anger on health-promoting behavior. In Model 1, among the sociodemographic and health-related characteristics, religion and exercise (which showed significant associations with health-promoting behaviors) were entered as independent variables into the regression model. Self-efficacy was additionally entered into Model 2, as was depression in Model 3 and trait anger, anger-in, anger-out, and anger control in Model 4.

Reviewer 2 Report

Dear author,

Thank you for your considerable work. Although the manuscript discusses a high-interest topic, some issues can be addressed to improve the quality and comprehensibility:

ABSTRACT

  1. The referee suggests specifying the range of age used in your sample in Ln 17.
  2. The referee suggests changing “Nurses need to pay attention to these factors to improve the health-promoting behaviors of Korean community-dwelling elderly women” with “It is necessary a nursing intervention strategy which pays attention to these factors to improve health-promoting behaviors of Korean community-dwelling elderly women”.

MATERIALS AND METHODS SECTION

  1. The authors should specify the range of age of the final sample used in this study, Ln 102.
  2. Ln 148-150, the authors explained before the meaning of anger-in, anger-out, and anger control in the Introduction section so that could be removed from this section.
  3. Ln 172, you should remove the comma “,” after “used”.
  4. Were the participants alone when they completed the questionnaires? I was just wondering how and when the researches read aloud the questionnaires to the participants with literacy difficulties. Maybe, the authors could include a phrase clarifying it (Ln 174-176).
  5. The referee suggests changing the order of the sentence in Ln 178-179, i.e. "participants were asked not to smoke or ingest caffeine-containing food or drinks such as coffee 30 minutes before the measurement. Blood pressure was measured using an automatic digital blood pressure OMRON HEM-181 7111(OMRON HEALTHCARE Co., LTD., JAPAN), while the participants were instructed to seat and raise their arm up to their chest. After the participants rested, two measurements were obtained at a 10-min interval".
  6. Ln 185, the referee suggests move “SPSS 23.0 was used for statistical analysis” to the final of the Statistical Analysis section.

RESULTS SECTION

  1. Ln 214-125, you could present the data in a different way, i.e. (129.67±SD and 122.34±SD, respectively; t = -1.98, p = 0.049).
  2. You could add a symbol such as “*” to the significative data in Tables 2 and 4 to make the results easier for the reader to understand.
  3. The authors should include in Table 5 the 4 Models performed.

DISCUSSION SECTION

  1. The authors should add a limitation paragraph at the final of the discussion section.
  2. In summary, although the work presented is interesting, it needs some small corrections in the Methods and Results sections in order to clarify and present better the information.

Author Response

The authors would like to thank the reviewers for carefully reviewing our manuscript and providing us Point and suggestions to improve the quality of the manuscript. We have tried our best to address all the reviewers’ concerns, and we will be happy to provide any further answers as needed. The revised sections are presented as red text.

Thanks again.

Response to Reviewer 2 Comments

 ABSTRACT

Point 1: The referee suggests specifying the range of age used in your sample in Ln 17.

Response: We added the range of age.

Self-report questionnaires were completed by 208 women aged 65 and older (age range 65 to 85) diagnosed with hypertension

 Point 2: The referee suggests changing “Nurses need to pay attention to these factors to improve the health-promoting behaviors of Korean community-dwelling elderly women” with “It is necessary a nursing intervention strategy which pays attention to these factors to improve health-promoting behaviors of Korean community-dwelling elderly women”.

Response: Thank you for this suggestion. We changed the wording to reflect your recommendation, with a slight modification for grammatical flow suggested by a native-speaker professional editor.

It is necessary to implement a nursing intervention strategy which pays attention to these factors to improve health-promoting behaviors of Korean community-dwelling elderly women.

MATERIALS AND METHODS SECTION

Point 3: The authors should specify the range of age of the final sample used in this study, Ln 102.

Response: We added the range of age.

Point 4: Ln 148-150, the authors explained before the meaning of anger-in, anger-out, and anger control in the Introduction section so that could be removed from this section.

Response: In accordance with your comment, we removed the meaning of anger-in, anger-out, and anger control.

Point 5: Ln 172, you should remove the comma “,” after “used”.

Response: We removed the comma “,” after “used”.

Point 6: Were the participants alone when they completed the questionnaires? I was just wondering how and when the researches read aloud the questionnaires to the participants with literacy difficulties. Maybe, the authors could include a phrase clarifying it (Ln 174-176).

Response: We added a further explanation of the data collection method.

The researchers and research assistants met participants in a consultation room provided by the community center. The researchers and research assistants remained with the participants and helped them fill out the questionnaires in a quiet and undisturbed space. Self-reported questionnaires were used and instructions were provided on how to complete them. Participants completed the questionnaire using a pen on their own if they were able to do so. A researcher then reviewed and collected the completed questionnaires. For participants who had literacy difficulties, mobility problems, or poor vision, a researcher read the questions aloud and a research assistant recorded the participants’ responses to the questions.

 Point 7: The referee suggests changing the order of the sentence in Ln 178-179, i.e. "participants were asked not to smoke or ingest caffeine-containing food or drinks such as coffee 30 minutes before the measurement. Blood pressure was measured using an automatic digital blood pressure OMRON HEM-181 7111(OMRON HEALTHCARE Co., LTD., JAPAN), while the participants were instructed to seat and raise their arm up to their chest. After the participants rested, two measurements were obtained at a 10-min interval".

Response: We changed the text to reflect your recommendation. Thank you for the suggestion.

participants were asked not to smoke or ingest caffeine-containing food or drinks such as coffee 30 minutes before the measurement. Blood pressure was measured using an automatic digital blood pressure reader (OMRON HEM-181 7111; Omron Healthcare Co., LTD., Japan), and the participants were requested to sit and raise their arm up to their chest for the measurements. After the participants rested, two measurements were obtained at a 10-minute interval.

Point 8: Ln 185, the referee suggests move “SPSS 23.0 was used for statistical analysis” to the final of the Statistical Analysis section.

Response: We moved this statement in accordance with your recommendation. Thank you for this helpful recommendation.

RESULTS SECTION

Point 9: Ln 214-125, you could present the data in a different way, i.e. (129.67±SD and 122.34±SD, respectively; t = -1.98, p = 0.049).

Response: We changed the corresponding text to reflect your recommendation. Thank you for the suggestion.

(129.67±21.07 and 122.34±22.07, respectively; t = -1.98, p = 0.049)

Point 10: You could add a symbol such as “*” to the significative data in Tables 2 and 4 to make the results easier for the reader to understand.

Response: We added such as “*” to the significative data in Tables 2 and 4 to align with your recommendation. Thank you for this helpful recommendation.

Point 11: The authors should include in Table 5 the 4 Models performed.

Response: We added the results for multicollinearity of the 4 models in Table 5.

DISCUSSION SECTION

Point 12: The authors should add a limitation paragraph at the final of the discussion section.

Response: We added a paragraph addressing some limitations to the end of the discussion section.

Point 13: In summary, although the work presented is interesting, it needs some small corrections in the Methods and Results sections in order to clarify and present better the information.

Response: Thank you for your considerate review. We revised the text to align with your recommendations to the best of our ability.

Reviewer 3 Report

The present study deals with various influences on high blood pressure in older Korean women.
The introduction and summary of the scientific basis are detailed and well-founded with sources from different decades. This is suitable to introduce the reader to the topic.
The methodology, as well as the procedure and the structure of the study is described very well and in a structured way and is easy to follow. Off remains the question how the questions were asked (with pen & paper or modile devices)? With pen & paper, how were transmission errors excluded during the evaluation?
The statistics are clearly presented in detail and are discussed in detail in the discussion and compared with the scientific state of the art. However, the test group of Korean women is directly generalized here. Here it would be necessary to make a short assessment of the influence of cultural and ethnic differences.

What's are G city's and N city's?

Author Response

The authors would like to thank the reviewers for carefully reviewing our manuscript and providing us Point and suggestions to improve the quality of the manuscript. We have tried our best to address all the reviewers’ concerns, and we will be happy to provide any further answers as needed. The revised sections are presented as red text.

Thanks again.

Response to Reviewer 3 Comments

Point 1: The present study deals with various influences on high blood pressure in older Korean women. The introduction and summary of the scientific basis are detailed and well-founded with sources from different decades. This is suitable to introduce the reader to the topic. The methodology, as well as the procedure and the structure of the study is described very well and in a structured way and is easy to follow.

Off remains the question how the questions were asked (with pen & paper or mobile devices)? With pen & paper, how were transmission errors excluded during the evaluation?

Response: In “2.3. Procedure,” we added a more detailed explanation of how the participants completed the questionnaires. We used self-reported pen-and-paper questionnaires. When the participants were completing the questionnaires, the researcher and the research assistant remained with them and provided explanations when the participants asked questions.  For participants who had literacy or comprehension difficulties, the researcher read the questions aloud and a research assistant recorded the participants’ responses to the questions.

The researchers and research assistants met participants in a consultation room provided by the community center. The researchers and research assistants remained with the participants and helped them fill out the questionnaires in a quiet and undisturbed space. Self-reported questionnaires were used and instructions were provided on how to complete them. Participants completed the questionnaire using a pen on their own if they were able to do so. A researcher then reviewed and collected the completed questionnaires. For participants who had literacy difficulties, mobility problems, or poor vision, a researcher read the questions aloud and a research assistant recorded the participants’ responses to the questions.

Point 2: The statistics are clearly presented in detail and are discussed in detail in the discussion and compared with the scientific state of the art. However, the test group of Korean women is directly generalized here. Here it would be necessary to make a short assessment of the influence of cultural and ethnic differences.

Response: Thank you for this important point. After discussion among the authors, we included a paragraph describing the limitations of this research, and noted that it is difficult to generalize the results from these participants.

This study has some limitations. First, it was difficult to generalize the results because the participants were Korean elderly women with hypertension from specific regions, and different results might occur depending on culture and ethnicity.

Point 3: What's are G city's and N city's?

 Response: These are the first letters of each city in which the study was conducted. These abbreviations were used to avert potential concerns regarding participant identifiability.
